# High Fidelity Pressure Wires Provide Accurate Validation of Non-Invasive Central Blood Pressure and Pulse Wave Velocity Measurements

**DOI:** 10.3390/biomedicines11041235

**Published:** 2023-04-21

**Authors:** Alessandro Scalia, Chadi Ghafari, Wivine Navarre, Philippe Delmotte, Rob Phillips, Stéphane Carlier

**Affiliations:** 1Department of Cardiology, Centre Hospitalier Universitaire Ambroise Paré, 7000 Mons, Belgium; 2Department of Cardiology, UMONS Research Institute for Health Sciences and Technology, University of Mons (UMONS), 7000 Mons, Belgium; 3Faculty of Medicine, Université Libre de Bruxelles, 1070 Bruxelles, Belgium; 4The School of Medicine, The University of Queensland, Brisbane 4072, Australia

**Keywords:** central blood pressure, cBP, hypertension, pulse wave velocity, PWV, invasive

## Abstract

Central blood pressure (cBP) is known to be a better predictor of the damage caused by hypertension in comparison with peripheral blood pressure. During cardiac catheterization, we measured cBP in the ascending aorta with a fluid-filled guiding catheter (FF) in 75 patients and with a high-fidelity micromanometer tipped wire (FFR) in 20 patients. The wire was withdrawn into the brachial artery and aorto-brachial pulse wave velocity (abPWV) was calculated from the length of the pullback and the time delay between the ascending aorta and the brachial artery pulse waves by gating to the R-wave of the ECG for both measurements. In 23 patients, a cuff was inflated around the calf and an aorta-tibial pulse wave velocity (atPWV) was calculated from the distance between the cuff around the leg and the axillary notch and the time delay between the ascending aorta and the tibial pulse waves. Brachial BP was measured non-invasively and cBP was estimated using a new suprasystolic oscillometric technology. The mean differences between invasively measured cBP by FFR and non-invasive estimation were −0.4 ± 5.7 mmHg and by FF 5.4 ± 9.4 mmHg in 52 patients. Diastolic and mean cBP were both overestimated by oscillometry, with mean differences of −8.9 ± 5.5 mmHg and −6.4 ± 5.1 mmHg compared with the FFR and −10.6 ± 6.3 mmHg and −5.9 ± 6.2 mmHg with the FF. Non-invasive systolic cBP compared accurately with the high-fidelity FFR measurements, demonstrating a low bias (≤5 mmHg) and high precision (SD ≤ 8 mmHg). These criteria were not met when using the FF measurements. Invasively derived average Ao-brachial abPWV was 7.0 ± 1.4 m/s and that of Ao-tibial atPWV was 9.1 ± 1.8 m/s. Non-invasively estimated PWV based on the reflected wave transit time did not correlate with abPWV or with atPWV. In conclusion, we demonstrate the advantages of a novel method of validation for non-invasive cBP monitoring devices using acknowledged gold standard FFR wire transducers and the possibility to easily measure PWV during coronary angiography with the impact of cardiovascular risk factors.

## 1. Introduction

Systemic hypertension (HTN) is a major cardiovascular risk factor contributing to overall mortality and morbidity, affecting 35–40% of the population and resulting in up to 10 million deaths worldwide each year [1,2].

The therapeutic reduction of HTN results in a decreased incidence of cardiovascular death, stroke, myocardial infarction, angina, and peripheral vascular disease [1].

Routine non-invasive brachial artery blood pressure (BP) measurement is a fundamental step in HTN diagnosis management and therapeutic guidance. Cuff-based automated oscillometric measurement of brachial BP (bBP) in the clinicians’ office is the most widespread method of BP monitoring. The 2018 European Society of Cardiology (ESC) guidelines define HTN as office systolic BP (BPsys) values ≥ 140 mmHg and/or diastolic BP (BPdia) values ≥ 90 mmHg [3], whereas the American College of Cardiology/American Heart Association (ACC/AHA) 2017 guidelines set the threshold at BPsys 130 mmHg and BPdia 80 mmHg [1]. The value of such guidelines in clinical practice is predicated on the accuracy of the technologies being used to measure BP and their accurate implementation. However, the assumption of devices’ measurement accuracy was challenged by a recent review of 2486 BP monitoring devices sold worldwide, with 73% having no validation at all [4]. The use of unvalidated BP devices may result in significant and unpredictable measurement errors, limit the utility of clinical BP monitoring, and limit the implementation of practice guidelines. Conversely, high-fidelity validation ensures the standardization of values and improved reliability and utilization of the technology.

Central (defined as aortic or carotid) blood pressure (cBP) measurements demonstrate superior clinical correlation with cardiovascular events (myocardial infarction, heart failure, acute coronary syndrome, or stroke) [5], as well as better correlation with left ventricular hypertrophy (LVH), carotid artery atherosclerosis, macular degeneration [6], and chronic renal failure [7] as compared with bBP. 

While arterial pressure varies constantly during the cardiac cycle, creating an arterial pressure wave, only systolic, mean, and diastolic pressures are routinely referenced. The pressure waveform also progressively changes along the arterial tree, and although mean and diastolic arterial pressures remain relatively constant, systolic pressure can increase by up to 40 mmHg between the aorta and the brachial artery [8]. This variability is due to the systolic pressure amplification in the peripheral vessels secondary to pressure wave reflection at branching points and a decrease in vessel caliber and progressive stiffening of the vessel walls, with both enhancing retrograde reflection of the pulse wave [9,10,11]. Peripheral stiffness is explained by the presence of a larger muscular component and a higher proportion of collagen in the distal arteries in comparison with the aorta, making them less distensible and more resistant, mainly in elderly hypertensive patients [6]. However, evidence suggests that the stiffness of the central arterial tree plays an important role in the increase in cBP with advancing age [12]. Vascular calcifications can occur both in the intima and media by an active cell-mediated process, resulting in transformation of vascular smooth muscle cells into osteoblast-type cells. In patients with rheumatoid arthritis, PWV and endothelial function seem to improve after anti-inflammatory treatment, suggesting that this condition is reversible [13]. Endothelial function plays a crucial role on the arterial stiffness by modulating levels of nitric oxide, and this effect is greater on the smaller arteries than on the aorta due to a higher percentage of smooth muscle cells in the media [14].

Antihypertensive drugs show different effects on cBP as compared with bBP [15]. Calcium channel antagonists, angiotensin converting enzyme (ACE) inhibitors, and angiotensin receptor blockers (ARB) have shown superior efficacy in reducing cBP compared with beta-blockers and thiazides [6], with amlodipine and indapamide showing superior efficacy to candesartan [15].

Hence, estimating cBP non-invasively in everyday practice is clinically attractive and might improve the diagnosis and management of HTN and clinical outcomes. Patients’ prognosis might also be improved through earlier and targeted screening of subpopulations at high risk of complications. To date, several cBP measuring devices are clinically available. Some use applanation tonometry of the radial artery and apply a mean transfer function to establish a central pressure curve [16,17]. 

We have shown the limited applicability of a generalized transfer function from radial recordings [18], while accessing the carotid artery gives a more proximate estimate of the aortic BP. Other devices extensively reviewed by Papaioannou et al. [19] use a brachial cuff-based oscillometric method.

Aortic pulse wave velocity (PWV) measurement, although difficult to achieve, was recently suggested by the European Society of Hypertension/European Society of Cardiology guidelines as the gold standard for the assessment of aortic stiffness [3,20]. Several large-scale studies and meta-analyses have shown that PWV is the single most important individual hemodynamic risk factor for the prediction of cardiovascular events [20,21,22], positively correlating with age, BMI, blood pressure, total cholesterol, homocysteine levels, and coronary CT calcium score [23]. Arterial stiffness is associated with left ventricular hypertrophy. Aging and cardiovascular risk factors lead to breaks in elastin fibers, the accumulation of collagen, fibrosis, inflammation, medial smooth muscle necrosis, and calcifications. For a while already, a consensus of experts recommended measuring arterial stiffness and central BP in order to evaluate the cardiovascular risks of patients, particularly in whom target organ damage had not been discovered by routine investigations [24].

Validation of new non-invasive cBP devices is most commonly performed by comparison with either a previously validated non-invasive method, which is suboptimal, or by comparison with invasive fluid-filled catheters (FF) [25]. FF catheters are like a harmonic oscillator limited by a low frequency bandwidth, resonating from their natural frequency and requiring adequate damping to avoid the overshoot of oscillations impacting their accuracy. 

High fidelity micromanometer tipped wires are used clinically to assess the fractional flow reserve (FFR) in coronary arteries and are an appealing alternative to FF, offering an invasive measurement of the PWV between the ascending aorta and the brachial artery. The PWV, and arterial bed stiffness in total, might explain reported high differences between measured and estimated cBP [26]. 

We sought to develop an accurate invasive reference method to validate new non-invasive devices measuring cBP and PWV using high-fidelity micromanometer tipped FFR wires for the measurement of cBP. The adoption of more precise and high-fidelity validation methods may improve the clinical practice of cBP monitoring, and improve the implementation of practice guidelines and clinical outcomes.

We report on the first application of FFR coronary wires for invasive cBP validation in the assessment of the Uscom BP+ (Uscom Ltd., Sydney, NSW, Australia) non-invasive cBP monitor. The BP+ measures cBP using a patent protected method based on brachial oscillometric pressure cuff measurements and applying a physics-based model of the left subclavian artery to brachial arterial branch to the low-frequency suprasystolic brachial artery pressure waveforms [27].

## 2. Materials and Methods

The study protocol was approved by the Institutional Review Board at both Centre Hospitalier Universitaire (CHU) Ambroise Paré—Mons, and Hôpital Erasme—Brussels, Belgium (references B406201732846 and B4062021000294) and adhered to the principles of the Declaration of Helsinki. Written informed consent was obtained from all patients before the examination.

Patients scheduled to undergo diagnostic cardiac catheterization and/or coronary angioplasty through the radial artery approach were enrolled (Figure 1). Patients who had acute coronary syndrome, known significant subclavian artery stenosis, arrhythmias, and aortic or mitral valvulopathies requiring surgery were excluded from the study.

Upon arrival to the catheterization laboratory, patients’ height and weight were noted. Left and right bBP were measured with the BP+ monitor to ensure minimal pressure difference between both arms and the absence of peripheral arterial disease. No sedative or premedication were used before or during the catheterization procedure. 

All routine medications were continued at the time of the procedure. After local injection of 2–3 mL of 1% lidocaine and successful placement of a 5Fr or 6Fr arterial sheath in the right or left radial artery, 5 mg of nicardipine were administered intra-arterially to prevent vasospasm during the procedure. Heparin (5000 U) was administered intravenously after insertion of the arterial catheter. The appropriate-size BP cuff was selected according to the manufacturer’s direction—24–32 cm or 32–42 cm—and was placed on the subject’s upper left arm with its lower edge 2.5 cm above the antecubital fossa.

At the end of the procedure, a NaCl 0.9% fluid-filled (FF) 5Fr or 6Fr Judkins Right catheter was stabilized in the ascending aorta, 2 cm above the aortic valve, under fluoroscopic guidance (*n* = 75). The distal end of the catheter was positioned away from the walls of the aorta to avoid measurement interference from kinetic energy transfer. The natural frequency and the damping of the FF pressure system was measured on the bounce of oscillations during a fast flush test (see Figure 2), as described by Gardner et al. [28]. Natural frequency was 1/time between peaks *A*1 and *A*2, as shown in Figure 2. Notably, Equation (1), reported in the appendix of Gardner’s paper and replicated by many authors [29], is flawed. The correct equation is presented in Gardner’s paper and should be used to compute the damping coefficient from the amplitude of the oscillations *A*1 and *A*2 as:(1)$$D=-\ln\left(\frac{A2}{A1}\right)/sqrt(\pi ²+\ln\left(\frac{A2}{A1}\right)²)$$

All non-invasive BP+ measurements were obtained with subjects in the supine position with the left arm positioned at mid-chest level. The simultaneous invasive central aortic pressure wave recordings from the Philips Hemolab and the automatic pressure measurement using the brachial cuff were repeated after diagnostic coronary angiography, with a total of 3 measurements for each participant. 

When clinically indicated (*n* = 20), fractional flow reserve was measured using a 0.014” wire including, at its tip, a high fidelity electronic pressure transducer (Comet wire, Boston Scientific, Marlborough, MA, USA) after additional Heparin (2500 U) was administered intravenously [30]. 

After coronary stenosis assessment, the guiding catheter was positioned in the ascending aorta and the FFR wire transducer was kept in the tip of the catheter. The wire-derived cBP was recorded contemporaneously with the non-invasive BP+ measurement. Recordings were repeated after pullback of the guiding catheter with the tip positioned in the brachial artery at the level of the mid-humerus. The distance of the pullback of the catheter between the ascending aorta and mid-humerus recording sites was measured with a tape measure.

The signals were digitalized on the Philips Hemolab at 100 Hz and exported as ASCII files for analysis on a custom-designed software written in MatLab^®^ (version r2017a, The MathWorks, Natick, MA, USA). Automatic batch analysis of all signals avoided intra-observer variations. Ten to twenty consecutive beats of the aortic and brachial pressure waves were averaged to limit the effect of physiological beat to beat variability (Figure 3). cBPsys and cBPdia were, respectively, identified as the peak- and end-diastolic values of the pulse pressure waveform, while an aortic pulse pressure (cPP) was computed as the difference between cBPsys and cBPdia. Aortic mean BP (cBPmean) was derived from the area under the averaged pressure curve and heart rate from the cardiac cycle length. Similar data were obtained from the averaged brachial pressure wave. The ARTERY Society Task Force validation guidelines recommend accuracy criteria for comparison of a non-invasive cBP measurement device with an invasive reference standard to achieve a mean difference ≤ 5 mmHg with a standard deviation (SD) ≤ 8 mmHg, and these standards were adopted for this study [31].

An aorto-brachial pulse wave velocity (abPWV) was calculated from the measured length of the pullback and the time delay between the ascending aorta and the brachial artery pulse wave foot estimated by the maximum upstroke of their second derivative, gating both measurements to the R-wave of the ECG [32]. 

On the digitalized signals, as well as on the iLab FFR monitor from Boston Scientific, we could visualize oscillations that were sometimes important on the FF catheter signals, associated with suboptimal damping as shown in Figure 4. However, we could verify that the foot of the FF and FFR pressure curves were very close to each other: the timing of the start of the pressure rise appears to be the least disturbed parameter. Following this observation, we recorded the ascending aortic BP wave with an FF catheter in 23 other patients undergoing coronary angiography, as described previously. Simultaneously, a cuff connected to an additional pressure transducer connected to the third channel of the hemodynamic Philips workstation of the cathlab was placed around the left leg, at mid-calf, and was inflated to record a tibial pulsatile BP wave. The distance between ascending aorta and tibial artery was estimated by the distance between the axillary and iliac crest, added with the distance between the iliac crest and mid-calf. The aorto-tibial atPWV was then calculated from this estimated length between AAo and mid-calf, and the time delay between the simultaneously recorded ascending aorta and tibial artery pulses and ECG. Ten to twenty consecutive beats of the aortic and tibial pressure waves were averaged after gating to the R wave of the ECG, using a dedicated software written in MatLab^®^.

Simultaneously, the Uscom BP+ was placed on the left arm to measure the reflected wave transit time (RWTT), which can estimate PWV using the jugulum to symphysis distance as PWV = 2 × jugulum-symphysis distance (m)/RWTT (s) [33]. The time necessary to record all BP measurements was only a couple of minutes added to the procedure, without additional X-ray exposure or contrast.

Bias and precision were determined using the Bland–Altman method [34] using SPSS^®^ Statistics version 23 software (IBM^®^, Armonk, NY, USA).

## 3. Results

### 3.1. Study Population

Among the 75 patients studied, 52 patients were recruited for the comparison between the cBP and the FF catheter, and 20 of them had an additional comparison with the FFR wire when a fractional flow reserve (FFR) measurement was clinically indicated. Subjects’ characteristics are shown in Table 1. The mean age was 62.5 ± 8.7 years, BMI 30.0 ± 5.8 kg/m^2^, with 67% male.

Aorto-tibial atPWV was measured in 23 additional patients who had a mean age of 60.4 ± 10.4 years and a BMI = 29.8 ± 4.3 kg/m^2^; 74% of them were male.

### 3.2. Invasive and Non-Invasive Central Blood Pressure Comparison

The mean differences between invasive FFR wire and non-invasive BP+ systolic, diastolic, and mean cBP were −0.4 ± 5.7 mmHg (CI 95%: −2.9; 2.1); −8.9 ± 5.5 mmHg (CI95%: −11.3; −6.5); and −6.4 ± 5.1 mmHg (CI95%: −8.6; −4.2), respectively (Figure 5). 

The mean differences between invasive FF measurements and non-invasive BP+ systolic, diastolic, and MAP cBP were 5.4 ± 9.4 mmHg (CI95%: 2.9; 8.0), −10.6 ± 6.3 mmHg (CI95%: −12.3; −8.9); and −5.9 ± 6.2 (CI95%: −7.6; −4.2), respectively. Linear regression comparing Uscom BP+ cBPsys and invasive FFR data show no statistical difference (*p* = 0.253), while comparison of Uscom BP+ cBPsys with FF measures demonstrates a significant difference (*p* = 0.041).

### 3.3. Invasive and Non-Invasive Brachial Blood Pressure Comparison

Invasive FFR measurements were compared with the oscillometric bBP measurements. The mean differences between bBP systolic and diastolic were −2.2 ± 9.8 mmHg (CI95%: −6.9; 2.7) and −7.9 ± 7.0 mmHg (CI95%: −11.3; 4.5), respectively (Figure 6). On the systolic chart, we can visually and statistically observe a linear regression (*p* = 0.005). This leads to an overestimation of bBPsys when BP is high, and an underestimation of bBPsys when the BP is low. Comparing the same data with invasive FF catheter showed, for systolic and diastolic blood pressure, −1.0 ± 11.1 mmHg (CI95%: −6.3; 4.3) and −10.8 ± 5.5 mmHg (CI95%: −13.5; −8.2).

### 3.4. Invasive FFR and FF Central Blood Pressure Comparison

In the 20 subjects where an FFR wire was used, we analyzed the correlation between FFR and FF cBP measurements (Figure 7). Measurements were well-correlated with an R^2^ of 0.815 and a Pearson correlation factor ρ = 0.903 (*p* < 0.001) for cBPsys. For cBPmean, R^2^ was 0.949 with the Pearson correlation factor ρ = 0.974 (*p* < 0.001) and for cBPdia, R^2^ was 0.936 with the Pearson correlation factor ρ = 0.967 (*p* < 0.001).

### 3.5. PWV Measurements

Aorto-brachial PWV (abPWV) was calculated in 20 patients. Two patients were excluded due to irregular RR intervals. Mean abPWV was found to be 7.2 ± 1.8 m/s. There was no statistically significant relationship between abPWV and cBPsys (*p* = 0.116), while there was a statistically significant overestimation of cBPmean when abPWV was higher (*p* = 0.021) (Figure 8). There was no relationship between age and abPWV in our rather small population of investigated patients (Pearson correlation coefficient ρ = 0.828).

The aorto-tibial PWV (atPWV) measured in the last 23 patients recruited in this study was 9.1 ± 1.8 m/s. Exploring whether measurements of atPWV might be simplified using the ECG instead of an invasive aortic BP curve, we measured the time between the R-wave of the QRS and the foot of the aortic blood pressure wave. This mean time delay was 66.7 ± 18.5 ms. On average, the simplified PWV computed using the time between the QRS and the foot of the tibial BP minus 66.7 ms was 9.3 ± 2.1 m/s, apparently close to the invasive gold standard atPWV. However, the R^2^ was 0.5 and the Pearson correlation factor was only 0.7, with unpredictable significant variations of up to 2 m/s in PWV, reflected by the significant (18.5 ms, ~25%) standard deviation of the mean QRS-aortic BP rise delay of 66.7 ms (Figure 9); this reflects the high variability in the electromechanical dissociation of the cardiac action.

### 3.6. PWV Measurements Derived from Reflected Wave Transit Time (RWTT)

Computed reflected wave transit time (RWTT) was obtained on the BP+ device and separated into RWTT peak and RWTT foot (Figure 10). Trachet et al. described a formula to estimate PWV using the RWTT parameter and the jugulum to symphysis distance as PWV = 2 × jugulum-symphysis distance (m)/RWTT (s). There was no correlation between abPWV and RWTT. For RWTTp, R^2^ was 0.019 with a Pearson correlation factor ρ = −0.137 (*p* = 0.59). For RWTTf, R^2^ was 0.013 with a Pearson correlation factor ρ = −0.114 (*p* = 0.65) (Figure 10).

RWTT was not better correlated with atPWV. For RWTTp, R^2^ was 0.003 with a Pearson correlation factor ρ = −0.052 (*p* = 0.817). For RWTTf, R^2^ was 0.202 with a Pearson correlation factor ρ = 0.45 (*p* = 0.36) (Figure 11).

### 3.7. Pulse Wave Velocity, Central Blood Pressure Estimation, and Cardiovascular Risk Factors

Patients with more than two cardiovascular risk factors (CVRF) demonstrated a trend for a 0.6 m/s higher PWV: the mean abPWV for patients with the ≤2 CVRF group was 6.8 m/s and for the ≥ 3 CVRF group was 7.4 m/s, but this difference was not statistically significant (*p* = 0.44). CVRF included obesity (BMI > 30 kg/m^2^), active smoking (or stopped less than two years), hypertension history (>130/80 mmHg), diabetes mellitus type 1 or 2, dyslipidemia, chronic renal failure (stage ≥ 4: MDRD ≤ 30 mL/min), and a personal history of STEMI or NSTEMI.

Regarding non-invasive cBP estimation, no significant differences were highlighted when comparing patients with ≤2 CVRF or ≥3 CVRF. The mean differences of cBPsys estimation were −2.5 ± 5.4 mmHg and 1.6 ± 5.8 mmHg, respectively (*p* = 0.076). The mean differences of cBPmean estimation were −6.7 ± 5.6 mmHg and −5.7 ± 5.3 mmHg, respectively (*p* = 0.128). The mean differences of cBPdia estimation were −7.9 ± 5.6 mmHg and −9.3 ± 6.0 mmHg, respectively (*p* = 0.518) (Figure 12).

## 4. Discussion

This study demonstrates the validation of a non-invasive oscillometric cBP device compared with measurements from a high-fidelity micromanometer tipped FFR catheter. The same method was applied to compare non-invasive oscillometric cBP measurements with the invasive, but less accurate, FF catheter.

Our data suggest the BP+ device and its algorithm accurately measured cBPsys, while there was a trend to overestimate cBPdia and cBPmean. Our findings concur with the meta-analysis from Papaioannou TG et al. [19], in which invasive cBPsys using an FF was compared with 11 different commercial devices in 808 study participants. Overall, the mean difference of cBPsys between methods was −4.5 mmHg, demonstrating an under-measurement of non-invasive cBP devices. Measurement of non-invasive cBPsys was found to be influenced by the calibration method, with better results obtained when invasive bBPsys was used to calibrate the non-invasive cBPsys (−1.1 mmHg versus −5.8 mmHg, non-invasively). However, comparison with that study is limited, as only 7 of the 22 studies used a supra-systolic wave analysis device, while the other studies used radial or carotid applanation tonometry.

Despite the high reliability of BP+ cBPsys measurements in our study, we demonstrate an over-measurement of cBPdia by approximately 5 mmHg, in line with the report of Gotzmann et al., who found a similar discrepancy comparing cBP measured invasively with a FF catheter and two non-invasive devices, one measuring the brachial volume displacement waveform with a cuff inflated at a pressure below brachial diastolic BP and the other one using an oscillometric recording of the brachial artery and applying the concept of pulse waves harmonics and Fourier analysis. This underestimation of cBPsys using an FF catheter mirrored our findings, in which the FF catheter significantly underestimated cBPsys. However, by using high-fidelity FFR wires we achieved the recommended cBPsys criteria for accuracy [25]. One can appreciate in Figure 2 and Figure 4 the limitations of the FF catheter: as a harmonic oscillator, it can sometimes have a suboptimal natural frequency and damping coefficient. In order to be able to reconstruct a minimum of eight harmonics of a pulse rate up to 180 bpm, the natural frequency of an arterial line should be at least 24 Hz. We measured on the oscillations during fast flush test on our recordings a natural frequency sometime as low as 7.5 Hz, like the one in Figure 2. An optimal damping is also crucial and should be between 0.64 and 0.7. On the same recording of Figure 2, it was only 0.2, limiting the overall frequency response of the arterial line (AL). Many factors will affect fluid-filled AL, such as the length and diameter of the tubing and the presence of unpredictable small air bubbles. Joffe et al. have reported the accuracy of AL using the flush test or stopcock test in critically ill children. They showed that only fifty percent of the AL were optimally damped, and that it was not predictable, with large variabilities even in the same patient on the same day [35]. On the contrary, the micromanometer of the FFR wire is not affected by these factors. The technical specifications of the Comet wire (Boston Scientific) that we used give a pressure accuracy of +/−3% or +/−3 mmHg, whichever is greater, a zero thermal effect of 0.3 mmHg/°C, and a frequency response > 25 Hz for an operating range between −45 and 300 mmHg.

Our study demonstrates that the Uscom BP+ cBPsys measurements compared with the high-fidelity FFR catheter measurements were accurate and met ARTERY criteria with a mean difference of −0.4 ± 5.7 mmHg, exceeding both recommended mean values ≤ 5 mmHg and SD ≤ 8 mmHg [31]. A previous study by Shoji et al. used a FFR validation of another non-invasive cBP device, the SphygmoCor XCEL (AtCor Medical, Sydney, Australia), and reported on 36 subjects a somewhat lesser agreement of cBPsys with a mean difference of 4.6 ± 9.9 mmHg, just falling short of the recommended ARTERY accuracy standards requiring an SD ≤ 8 mmHg [36].

The first and qualifying step for the measurement of an accurate cBP is the accurate measurement of a bBP under reproducible conditions. While the non-invasive mean bBPsys was accurate despite a wide SD measurement, the bBPdia and bBPmean did not achieve the level of accuracy recommended by the ARTERY according to SD and the mean criteria. This finding prompted the manufacturer to review the calculations for diastolic and mean bBP, and additional subjects are currently being recruited for this ongoing validation.

Invasive and non-invasive cBPsys linear regression analysis with the FF catheter showed a positive trend when BP increased (*p* = 0.041), indicating an underestimation of the Uscom BP+ device as cBPsys increased, while cBPdia and cBPmean were found to be unaffected. A finding which may have been associated with inaccuracies in the FF catheter is that this trend was not found when using the pressure of the FFR wire as the reference. Moreover, we did not find an influence of the number of cardiovascular risk factors on the accuracy in the estimation of cBPsys (Figure 12).

We further demonstrate that the FF catheter shows artefactual alterations to the pressure wave with an increase in the systolic peak, as compared with the FFR catheter, and a potential cause for inaccuracy (Figure 5). 

While reference values for PWV are defined for carotid-femoral or brachial-ankle methods [3,37,38], there remains no consensus on reference values for aorto-brachial PWV values. Our invasive PWV values cannot be compared with published values, but remain an alternate and novel method for measuring PWV.

To the best of our knowledge, no other study has investigated the influence of PWV on cBP measurement. We found no relationship between PWV and cBPsys. However, when PWV was higher, there was a tendency to overestimate cBPmean with an error not exceeding 10 mmHg and not considered excessive when compared with other errors of BP measurement [39].

Nam et al. reported a positive correlation between age and brachial-ankle PWV [23]. In our study, no relationship was established between age and aorto-brachial PWV, which may potentially be explained by a small population sample. Of consideration is that the measurement of PWV is sensitive to any subclavian artery stenosis, and 14% of the subjects in our study suffered from arterial disease in the lower extremity, a condition likely associated with undiagnosed arterial disease of the upper extremity.

Estimating PWV non-invasively is clinically attractive, but any simplified approach needs to hold accurate estimates against a straightforward invasively measured reference PWV. The reflected wave transit time (RWTT) method appears to fail this target in our validation study. Our validation study confirms issues raised from numerical simulations in a model of the arterial tree [33]. One of the challenges might include the assessment of the reflection site, supposed to be aorto-iliac bifurcation. Conversely, an oversimplistic approach that would replace the assessment of the timing of the foot of the ascending aorta blood pressure by the R-wave of the ECG is also not an alternative. We demonstrated important and unpredictable large variations, with estimated secondary errors in the estimation of the PWV of up to more than 2 m/s.

## 5. Limitations

The normal respiratory variation in BP waves complicates accurate comparison of contemporaneous data samples and may limit the comparison of results (Figure 13).

Measurement of the PWV is derived from the distance of the pull back of the guide wire. As the catheter is introduced via the aorta into the sinus of Valsalva, the normal anatomical architecture of the aortic arch and any degenerative tortuosity may alter the pullback measurement and lead to an overestimation of PWV.

Methodologically, this study compares the accuracy of FF catheters and FFR wires for the calibration and validation of non-invasive cBP measurements. While FFR is acknowledged as a superior measure and this novel study demonstrated a superior agreement of Uscom BP+ cBP measures and FFR values, this in isolation does not disprove the accuracy of FF. Yet, we identified many factors that may affect the frequency response of a FF system and may limit its use for non-invasive BP device validation. 

The use of an FFR wire micromanometer is likely to be more accurate, and therefore requires fewer observations and patients for validation.

## 6. Conclusions

There is an increasing interest in the non-invasive measurement of cBP due to its better prediction of cardiovascular events and the potential benefits of its routine adoption in the clinical environment [27]. Accuracy is, consequently, critical for the widespread adoption of these non-invasive technologies. The implementation of high-fidelity validation of cBP monitors may lead to more accurate cBP measurements and an increased adoption of practice guidelines, leading to a reduction in the cardiovascular complications of HTN and a more widespread adoption of these emerging non-invasive cBP measurement technologies. This study demonstrates a novel validation method for the calibration of non-invasive cBP monitoring devices using the acknowledged gold standard high-fidelity FFR wire catheter. This study demonstrates that the FFR wire method is significantly different to the FF method and, further, that the Uscom BP+ accurately and non-invasively measures cBPsys without interference from the PWV and cardiovascular risk factors. Further studies are needed to refine our understanding of the relationship between PWV and cardiovascular risk factors and the clinical benefits of systematically assessing PWV during a cardiac catheterization. More outcome data will prove the benefits of using cBP to guide HTN treatment.

Finally, we demonstrated the difficulties of estimating PWV without invasive measurements, using a default value for aortic BP as well as PWV derived from the RWTT parameter. However, we described an easy and reliable method to measure invasive PWV, which could be used for all patients presenting for coronary angiography.

## Figures and Tables

**Figure 1 biomedicines-11-01235-f001:**
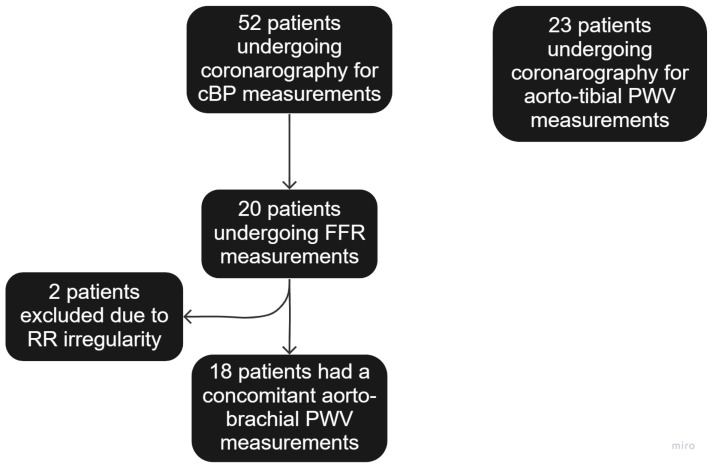
Flowchart of patients’ recruitment. Two patients with an FFR wire were excluded due to RR irregularity preventing PWV measurement.

**Figure 2 biomedicines-11-01235-f002:**
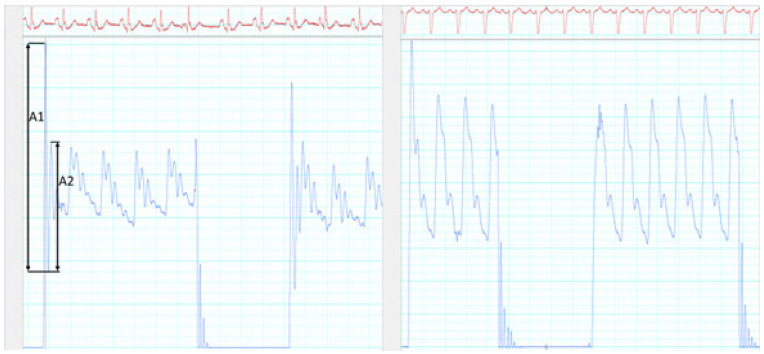
Measurement of the natural frequency and the damping coefficient during a fast flush test on the fluid-filled arterial line on the **left** panel. The **right** panel shows a much better frequency response and optimal damping without oscillation on the recording with an FFR wire in another patient. A1 is the amplitude of the first peak, A2 the amplitude of the second one.

**Figure 3 biomedicines-11-01235-f003:**
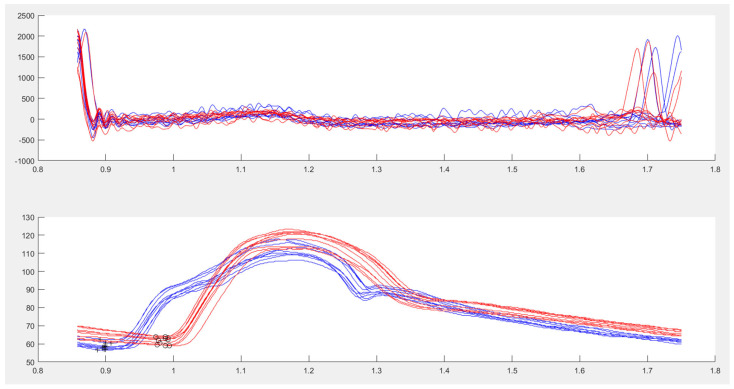
Measurement method to calculate aorto-brachial pulse wave velocity. **Upper** panel: superposition of R-wave of the electrocardiogram for both measurements. **Lower** panel: superposition of invasive aortic (blue) and brachial (red) pulse pressure waves, gated to the R-wave of the electrocardiogram and the foot of each BP rise (+ for aortic, o for brachial).

**Figure 4 biomedicines-11-01235-f004:**
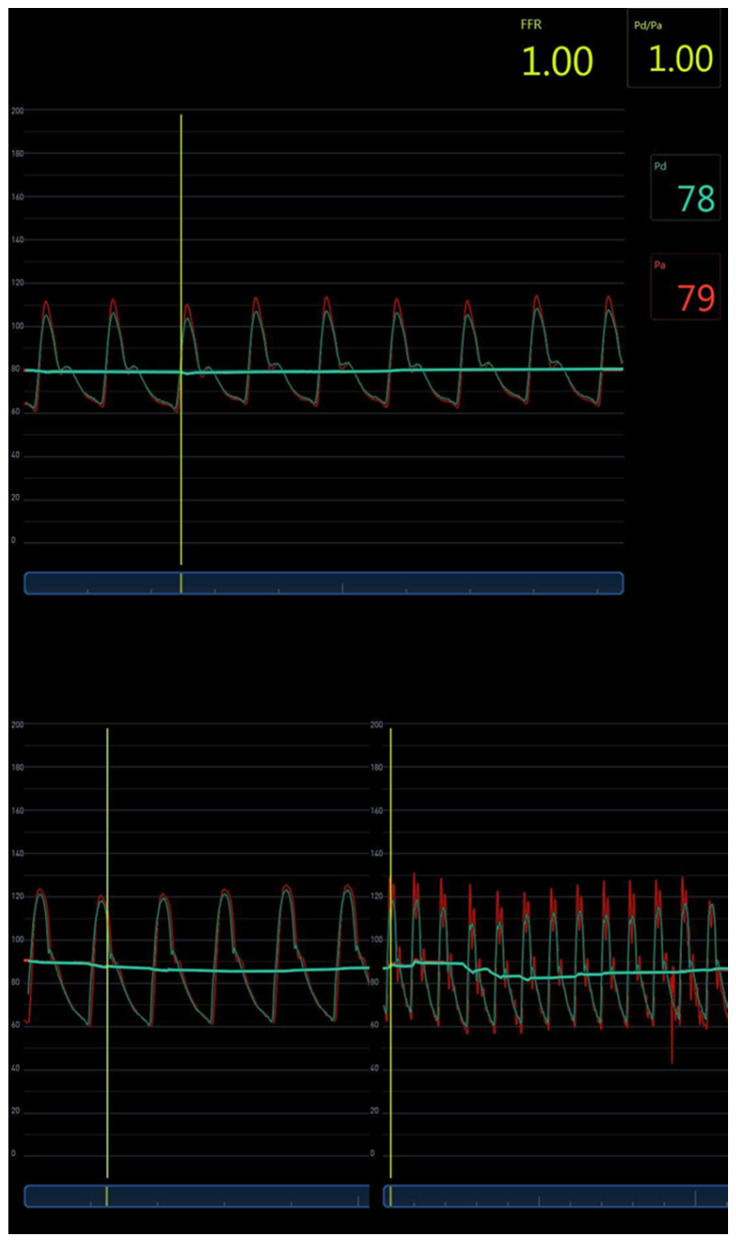
Screenshots of the recordings with the iLab (Boston Scientific) of the FFR pressure wire (Pd) in blue and the fluid-filled pressure (Pa) in red in 3 different patients. The pressure wire was equalized in the guiding catheter left in the ascending aorta, so that its mean value equals the mean fluid-filled pressure, resulting in an FFR value of 1. The lower right panel demonstrates a clearly underdamped system with many oscillation artefacts, while the timing of the feet of the Pa and Pd pressure waves is the same for these patients.

**Figure 5 biomedicines-11-01235-f005:**
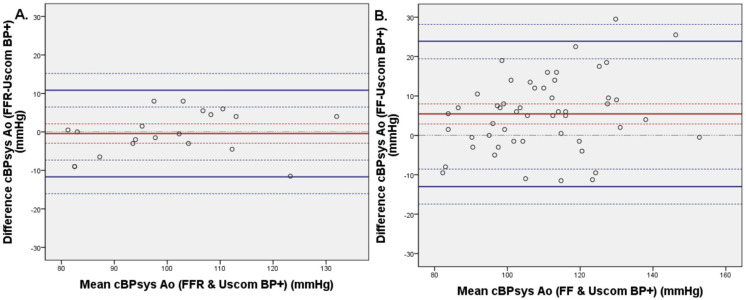
Bland–Altmann plots of systolic (**A**,**B**), mean (**C**,**D**), and diastolic (**E**,**F**) central blood pressure comparing non-invasive blood pressure (Uscom BP+ device) with invasive central blood pressure measured with FFR wire on the left side (*n* = 20) and with FF catheter on the right side (*n* = 52).

**Figure 6 biomedicines-11-01235-f006:**
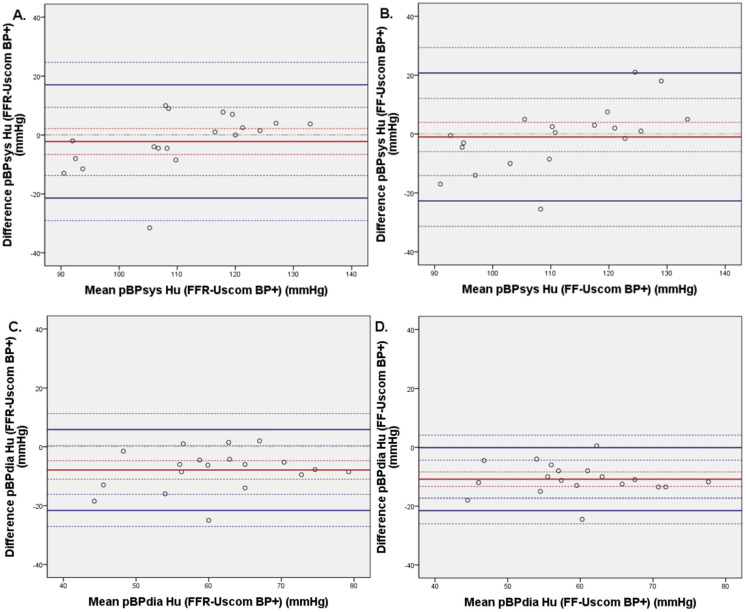
Bland–Altmann plots of systolic (**A**,**B**) and diastolic (**C**,**D**) brachial blood pressure comparing non-invasive blood pressure (Uscom BP+ device) to invasive brachial blood pressure (with FFR wire on the left side and with FF catheter on the right side) (*n* = 20).

**Figure 7 biomedicines-11-01235-f007:**
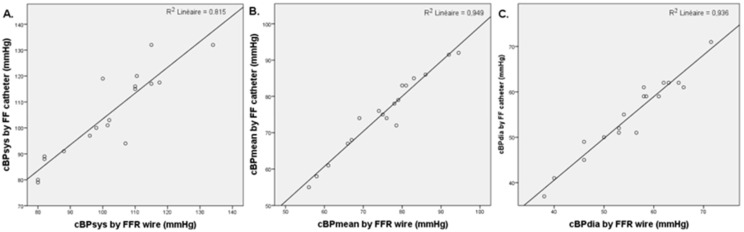
Correlation plot of cBPsys (**A**), cBPmean (**B**), and cBPdia (**C**) between the high-fidelity measurements and the FF measurements.

**Figure 8 biomedicines-11-01235-f008:**
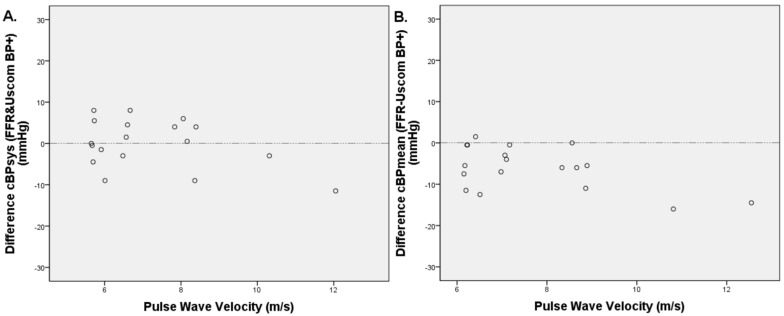
Dispersion graph between aorto-brachial pulse wave velocity and the difference of invasive and non-invasive cBPsys (**A**) and cBPmean (**B**) (*n* = 18).

**Figure 9 biomedicines-11-01235-f009:**
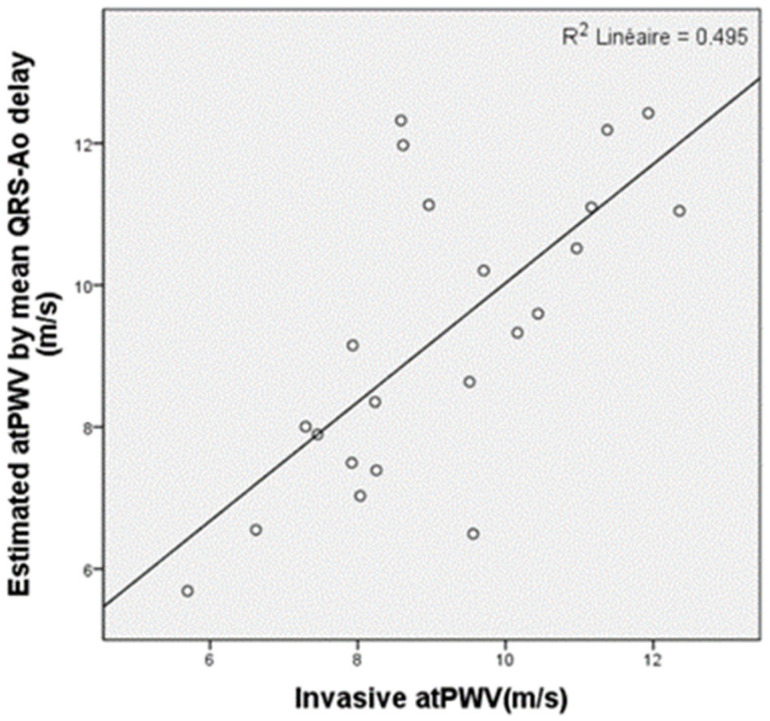
Correlation plot between invasive aorto-tibial PWV and estimated PWV using the mean time delay between the R-waves of the QRS and the aortic blood pressure wave.

**Figure 10 biomedicines-11-01235-f010:**
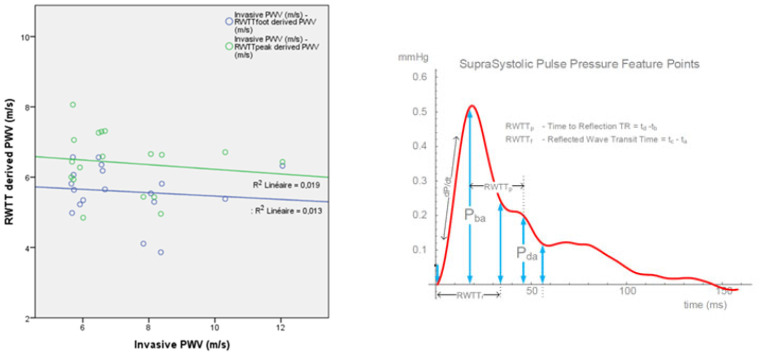
On the **left**, correlation plot between the invasive aorto-brachial pulse wave velocity (PWV) and the reflected wave transit time (RWTT) derived PWV, showing a lack of correlation between both parameters. On the **right**, representation of a suprasystolic brachial pulse pressure wave and definition of RWTT peak (RWTTp) and RWTT foot (RWTTf).

**Figure 11 biomedicines-11-01235-f011:**
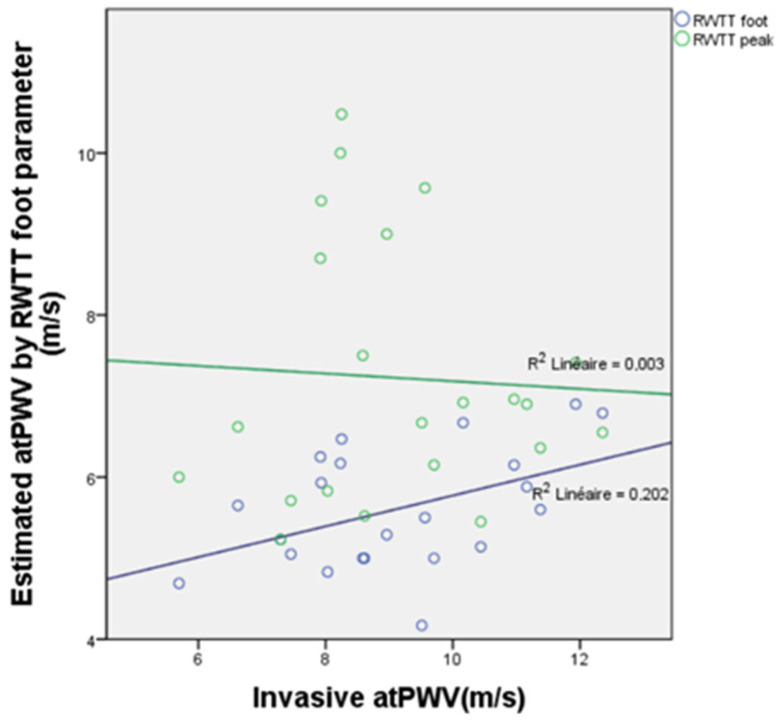
Correlation plot between the invasive aorto-tibial PWV and the RWTT-derived PWV, showing a lack of correlation between both parameters.

**Figure 12 biomedicines-11-01235-f012:**
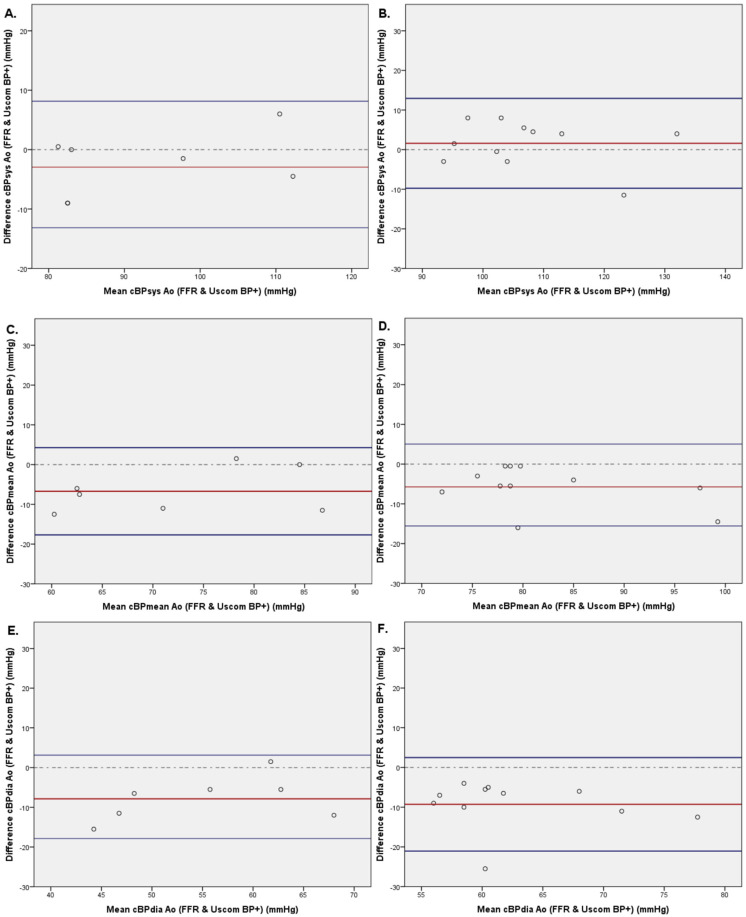
Bland–Altmann plots of systolic (**A**,**B**), mean (**C**,**D**), and diastolic (**E**,**F**) central blood pressure comparing non-invasive blood pressure (Uscom BP+ device) to invasive central blood pressure measured with FFR wire separation on patients with ≤2 cardiovascular risk factors (CVRF) (**A**,**C**,**E**) or ≥3 CVRF (**B**,**D**,**F**).

**Figure 13 biomedicines-11-01235-f013:**
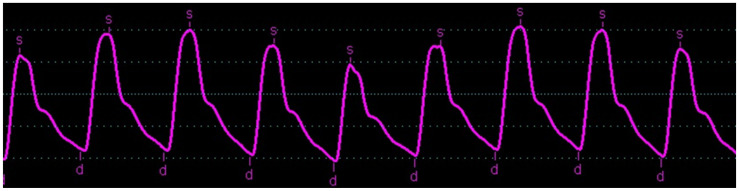
Illustration of the normal respiratory variation of the central blood pressure curve as intrathoracic filling pressures flux with inspiration and expiration, and so potentially limiting any accuracy of methods comparison. S represent systolic blood pressure and d represent diastolic blood pressure.

**Table 1 biomedicines-11-01235-t001:** Clinical characteristics of the patients included. CAD = coronary artery disease. CKD = chronic kidney disease. LVH = left ventricular hypertrophy.

Characteristic	cBP Comparison and n = 52 (%)	atPWV n = 23 (%)
Smoking history ◦ Non smoker ◦ Current smoker	26 *(50)* 12 *(23)*	17 *(74)* 6 *(26)*
Hypertension	34 *(65)*	12 *(52)*
Diabetes mellitus	24 *(46)*	9 *(39)*
Dyslipidemiae	50 *(96)*	16 *(70)*
CAD history	19 *(37)*	
CKD	17 *(33)*	
LVH	11 *(21)*	

## Data Availability

Not applicable.

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
