# Peer review of "High Fidelity Pressure Wires Provide Accurate Validation of Non-Invasive Central Blood Pressure and Pulse Wave Velocity Measurements"

_biomedicines, 2023, doi:10.3390/biomedicines11041235_

Round 1
Reviewer 1 Report
Dear authors, dear editors,
Calier's work demonstrates the possibilities of invasive validation with regard to non-invasive measurements of central blood pressure parameters and pulse wave velocity. The work has some methodological strengths, but unfortunately also some open points that should be addressed in a revision.
In the section “introduction” (lines 56-62), the endothelial dysfunction and the further pathogenesis of arterial vascular stiffness should be presented further and more differentiated, in addition to the change with a large proportion of collagen.
In the materials and methods section, the recommendation would be a flowchart for inclusion and exclusion from the subjects and which measurements which patients receive according to the consort statement (line 107-110).
In the results section, it would be very valuable if a table with the possible comedications were added (Table 1 - rows 163-165).
In addition, the question arises why PWV has only been measured in 18 patients, this would require a further graph or integration into the flowchart and additional comments in the area of inclusion and exclusion criteria.
Author Response
Dear Reviewer,
Thank you for your review which will improve the quality of our manuscript. You proposed:
Calier's work demonstrates the possibilities of invasive validation with regard to non-invasive measurements of central blood pressure parameters and pulse wave velocity. The work has some methodological strengths, but unfortunately also some open points that should be addressed in a revision.
In the section “introduction” (lines 56-62), the endothelial dysfunction and the further pathogenesis of arterial vascular stiffness should be presented further and more differentiated, in addition to the change with a large proportion of collagen.
As proposed, we added the impact of endothelial function in the introduction section page 2:
Vascular calcifications can occur both in the intima and media by an active cell-mediated process resulting in transformation of vascular smooth muscle cells into osteoblast-type cells. In patients with rheumatoid arthritis, PWV and endothelial function seems to improve after anti-inflammatory treatment suggesting that this condition is reversible [12]. Endothelial function plays a crucial role on the arterial stiffness by modulating levels of nitric oxide, and this effect is greater on the smaller arteries than on the aorta due to a higher percentage of smooth muscle cells in the media [13].
In the materials and methods section, the recommendation would be a flowchart for inclusion and exclusion from the subjects and which measurements which patients receive according to the consort statement (line 107-110).
In addition, the question arises why PWV has only been measured in 18 patients, this would require a further graph or integration into the flowchart and additional comments in the area of inclusion and exclusion criteria.
We have added a flowchart explaining our subject's population on page 3, and clarified why we had 18 PWV measurements instead of 20. Of note, we have added data about a second cohort of patients where we have recorded simultaneously a fluid-filled aortic BP and the pressure of a cuff inflated around the calf in order to derive an alternative PWV taking a longer aortic path, more comparable to the PWV derived from the tonometric approach based on the recording of carotid and femoral pulses. We have of course added our colleague, Dr Wivine Navarre as co-author, who performed these measurements. This was to partly address the editorial request to expand our manuscript to +4000 words. We also added more discussion on the issues of using FF catheters with additional figures also.
On page 4:
The natural frequency and the damping of the FF pressure system was measured on the bounce of oscillations during a fast flush test (see Figure 2) as described by Gardner et al. [26]. Natural frequency was 1 / time between peaks A1 and A2 as shown on figure 2. Of note, the equation (3) reported in the appendix of Gardner’s paper, and replicated by many authors [27] is flawed. The correct equation is presented on the graph of figure 10 in Gardner’s paper and should be used to compute the damping coefficient from the amplitude of the oscillations A1 and A2 as:
)
On page 6:
On the digitalized signals, as well as on the iLab FFR monitor from Boston Scientific we could visualize oscillations that were sometimes important on the FF catheter signals, and associated with suboptimal damping as shown on figure 4. However, we could verify that the foot of the FF and FFR pressure curves were very close to each other: the timing of the start of the pressure rise appears to be the least disturbed parameter. Following this observation, we recorded the ascending aortic BP wave with a FF catheter in 23 other patients undergoing coronary angiography as described previously. Simultaneously, a cuff connected to an additional pressure transducer connected to the third channel of the haemodynamic Philips workstation of the cathlab was placed around the left leg, at mid-calf, and was inflated to record a tibial pulsatile BP wave. The distance between ascending aorta and tibial artery was estimated by the distance between the axillary and iliac crest, added with the distance between iliac crest and mid-calf. The aorto-tibial atPWV was then calculated from this estimated length between AAo and mid-calf, and the time delay between the simultaneously recorded ascending aorta and tibial artery pulses and ECG. Ten to twenty consecutive beats of the aortic and tibial pressure waves were averaged after gating to the R wave of the ECG, using a dedicated software written in MatLab®.
On page 15:
The aorto-tibial PWV (atPWV) measured in the last 23 patients recruited in this study was 9.1 ± 1.8 m/sec. Exploring whether measurements of atPWV might be simplified using the ECG instead of an invasive aortic BP curve, we measured the time between the R-wave of the QRS and the foot of the aortic blood pressure wave. This mean time delay was 66.7 ± 18.5 ms. On average, the simplified PWV computed using the time between the QRS and the foot of the tibial BP minus 66.7 ms was 9.3 ± 2.1 m/sec, apparently close to the invasive gold-standard atPWV. However, the R² was 0.5 and the Pearson correlation factor was only 0.7 with unpredictable significant variations of up to 2 m/s in PWV, reflected by the significant (18.5 ms, ~25%) standard deviation of the mean QRS-aortic BP rise delay of 66.7 ms (Figure 7).
In the discussion
One can appreciate on figures 2 and 4 the limitation of the FF catheter, a harmonic oscillator that can sometime have a suboptimal natural frequency and damping coefficient. To be able to reconstruct a minimum of 8 harmonics of a pulse rate up to 180 bpm, the natural frequency of an arterial line should be at least 24 Hz. We measured on the oscillations during fast flush test on our recordings, a natural frequency as low as 7.5 Hz, like the one in figure 2. An optimal damping is also crucial and should be between 0.64 and 0.7. On the same recording, it was only 0.2, limiting the overall frequency response. Many factors will affect FF arterial line, such as the length and diameter of the tubing and the presence of unpredictable small air bubbles. On the contrary, the micromanometer of the FFR wire is not affected by these factors and has typically a frequency range >25 Hz.
In the results section, it would be very valuable if a table with the possible comedications were added.
Unfortunately, we did not record all medications and we cannot include the requested comedications table.
Reviewer 2 Report
Dear authors,
I want to thank you for the interesting article High fidelity pressure wires provide accurate validation of 2 non-invasive central blood pressure and pulse wave velocity 3 measurements. Actually, I don't have any significant comments, rather minor adjustments and maybe a thought/s to think about.
Abstract and introduction - without comments
Line 102 - an illustrative picture of the engagement and measurement in the study would be appropriate
Line 104 – permit number and ethical commission number are missing
Line 179-181; 194-196; 206; 204 - it would be appropriate to insert basic statistical descriptions into the pictures
At the same time, improve the description in the images, unreadable at the given DPI
It wouldn't hurt, where there are a lot of pictures, to mark them (eg A, B, C, ...) and put the given description in the description of the pictures
Add the number of test subjects n= to the pictures
Line 222 - typo FFF - FFR
Line 227 - adjust the formulas that go through the correlation lines, put them further away or in the corner of the picture
Line 330 – punctuation error at the end of the sentence
Line 381 - badly written citation - error/typo in the name
Since I have been studying the differences between many (easily accessible) BP measurement methods for a long time, I cannot help but ask to add a conclusion. It is about clearly establishing visions, what needs to be done in the next step and in which direction scientific and professional attention should be directed. I'm commenting on this because I myself am going to connect several cBP measurements with non-invasive measurements in the near future. I myself am curious as to how individual aspects of the test affect BP, PWV and other parameters. A clearly established vision will help not to repeat the measurements of others, but to know how to involve other possibilities to expand knowledge.
Author Response
Dear Reviewer,
Thank you very much for your careful review and very kind comments at the end. We have reworked our manuscript accordingly that we feel much improved with your corrections and suggestions.
I want to thank you for the interesting article High fidelity pressure wires provide accurate validation of 2 non-invasive central blood pressure and pulse wave velocity 3 measurements. Actually, I don't have any significant comments, rather minor adjustments and maybe a thought/s to think about.
Abstract and introduction - without comments
Line 102 - an illustrative picture of the engagement and measurement in the study would be appropriate
Line 104 – permit number and ethical commission number are missing
Line 179-181; 194-196; 206; 204 - it would be appropriate to insert basic statistical descriptions into the pictures
At the same time, improve the description in the images, unreadable at the given DPI
It wouldn't hurt, where there are a lot of pictures, to mark them (eg A, B, C, ...) and put the given description in the description of the pictures
Add the number of test subjects n= to the pictures
Line 222 - typo FFF - FFR
Line 227 - adjust the formulas that go through the correlation lines, put them further away or in the corner of the picture
Line 330 – punctuation error at the end of the sentence
Line 381 - badly written citation - error/typo in the name
Since I have been studying the differences between many (easily accessible) BP measurement methods for a long time, I cannot help but ask to add a conclusion. It is about clearly establishing visions, what needs to be done in the next step and in which direction scientific and professional attention should be directed. I'm commenting on this because I myself am going to connect several cBP measurements with non-invasive measurements in the near future. I myself am curious as to how individual aspects of the test affect BP, PWV and other parameters. A clearly established vision will help not to repeat the measurements of others, but to know how to involve other possibilities to expand knowledge.
We have corrected all the minor comments as proposed.
We have added the reference of our IRB application (references B406201732846 and B4062021000294) and a flowchart explaining our subject's population on page 3.
We also improved the visibility of all legends and descriptions in the images.
We can only agree with your vision of the future. To be more practical, we have added figures and a discussion on the physical limitations of fluid filled catheters. We calculated from measurements of fast flushed that we performed, but did not report on initially, the natural frequency and damping of the FF system and we added a discussion, as requested, where one should pay attention when validating with invasive recordings a cBP device.
Sincerely yours,
Stéphane G. Carlier
Round 2
Reviewer 1 Report
Dear editor, dear authors, the work has been significantly improved by the revision. Unfortunately, aspects raised regarding cardiovascular risk factors have not been addressed, but this should be addressed, but this is only a small aspect.
Author Response
view file uploaded
